# ICA1L Is Associated with Small Vessel Disease: A Proteome-Wide Association Study in Small Vessel Stroke and Intracerebral Haemorrhage

**DOI:** 10.3390/ijms23063161

**Published:** 2022-03-15

**Authors:** Natalia Cullell, Cristina Gallego-Fábrega, Jara Cárcel-Márquez, Elena Muiño, Laia Llucià-Carol, Miquel Lledós, Jesús M. Martín-Campos, Jessica Molina, Laura Casas, Marta Almeria, Israel Fernández-Cadenas, Jerzy Krupinski

**Affiliations:** 1Stroke Pharmacogenomics and Genetics Group, Biomedical Research Institute Sant Pau, 08041 Barcelona, Spain; ncullell@mutuaterrassa.cat (N.C.); cristina.gallego.fabrega@gmail.com (C.G.-F.); jara.carcel@gmail.com (J.C.-M.); elena.muinho@gmail.com (E.M.); laialluciacarol@gmail.com (L.L.-C.); miquel.lledos@gmail.com (M.L.); jmartinca@santpau.cat (J.M.M.-C.); 2Neurology Department, Hospital Universitari MútuaTerrassa, 08221 Terrassa, Spain; jessicamolina@mutuaterrassa.cat (J.M.); lcasas@mutuaterrassa.cat (L.C.); malmeria@mutuaterrassa.cat (M.A.); 3Fundacio Docència i Recerca MutuaTerrassa, 08221 Terrassa, Spain; 4Department of Medicine, Universitat de Barcelona, 08036 Barcelona, Spain; 5Department of Medicine, Universitat Autònoma de Barcelona, 08193 Barcelona, Spain

**Keywords:** small vessel disease, small vessel stroke, intracerebral haemorrhage, PWAS, proteome-wide association study, genome-wide association study (GWAS)

## Abstract

Small vessel strokes (SVS) and intracerebral haemorrhages (ICH) are acute outcomes of cerebral small vessel disease (SVD). Genetic studies combining both phenotypes have identified three loci associated with both traits. However, the genetic cis-regulation at the protein level associated with SVD has not been studied before. We performed a proteome-wide association study (PWAS) using FUSION to integrate a genome-wide association study (GWAS) and brain proteomic data to discover the common mechanisms regulating both SVS and ICH. Dorsolateral prefrontal cortex (dPFC) brain proteomes from the ROS/MAP study (N = 376 subjects and 1443 proteins) and the summary statistics for the SVS GWAS from the MEGASTROKE study (N = 237,511) and multi-trait analysis of GWAS (MTAG)-ICH–SVS from Chung et al. (N = 240,269) were selected. We performed PWAS and then a co-localization analysis with COLOC. The significant and nominal results were validated using a replication dPFC proteome (N = 152). The replicated results (*q*-value < 0.05) were further investigated for the causality relationship using summary data-based Mendelian randomization (SMR). One protein (ICA1L) was significantly associated with SVS (z-score = −4.42 and *p*-value = 9.6 × 10^−6^) and non-lobar ICH (z-score = −4.8 and *p*-value = 1.58 × 10^−6^) in the discovery PWAS, with a high co-localization posterior probability of 4. In the validation PWAS, ICA1L remained significantly associated with both traits. The SMR results for ICA1L indicated a causal association of protein expression levels in the brain with SVS (*p*-value = 3.66 × 10^−5^) and non-lobar ICH (*p*-value = 1.81 × 10^−5^). Our results show that the association of *ICA1L* with SVS and non-lobar ICH is conditioned by the cis-regulation of its protein levels in the brain.

## 1. Introduction

Cerebral small vessel disease (SVD) is a chronic and progressive heterogeneous disorder affecting small arteries, arterioles, venules, and capillaries in the brain [1,2]. Among the different outcomes of SVD, small vessel stroke (SVS) and spontaneous intracerebral haemorrhage (ICH) are the most acute and devastating manifestations [3,4,5]. One-quarter of all ischemic strokes (IS) are SVS [6], and ICH is the most common cause of haemorrhagic stroke [7]. The incidence of SVD increases with age, and it is, thus, expected that with the increase in life expectancy, the frequency of SVD manifestations will also increase in following years [1,3].

SVD causes parenchyma lesions, such as white matter hyperintensities (WMHs), enlarged perivascular spaces, lacunes, small subcortical infarcts, cerebral microbleeds, and brain atrophy, among others [1,3]. These parenchyma lesions are used as surrogate markers of SVD and can be quantified using brain imaging techniques. The extent of these lesions predicts poor outcomes after ICH [8] and SVS [9]. The two major pathophysiological mechanisms beyond SVD are arteriosclerosis and cerebral amyloid angiopathy (CAA) [1]. Cardiovascular risk factors, mainly hypertension, are associated with arteriosclerotic SVD. It can cause the rupture or occlusion of small arteries and arterioles, producing non-lobar (or deep) ICH and SVS, respectively [10].

There is no effective treatment for SVD, apart from controlling cardiovascular risk factors [4]. Genome-wide association studies (GWAS) have been helpful in identifying genes involved in SVD, analysing magnetic resonance image (MRI) SVD surrogate markers [11,12,13,14,15,16,17,18,19,20,21], and, moreover, specifically studying SVS and ICH.

Two GWAS have been performed with ICH patients. The first study identified a risk locus for non-lobar ICH in 1q22, overlapping with *PMF1* and *SLC25A44* genes [5,11,12]. The second performed a meta-analysis that allowed for the identification of *APOE* associated with lobar ICH [4]. In order to improve the statistical power of these studies, Chung et al. applied a novel strategy, using a multi-trait analysis of GWAS (MTAG) and combining genotype data from two related phenotypes, ICH and SVS. This approach permitted the identification of two novel associations in 2q33 (*ICA1L*) and 13q34 (*COL4A2* and *COL4A1*) associated with non-lobar ICH [4]. Interestingly, these loci presented a nominal association (*p*-value < 5 × 10^−3^) with non-lobar ICH and SVS in the corresponding GWAS but not at a genome-wide level [4]. Three different GWAS in SVS have identified significant associations with this phenotype in 16q24.2 (*ZCCHC14*) [6], 2q33.2 (*ICA1L* and *WDR12*) [22], *ULK4*; *SPI1-SLC39A13-PSMC3-RAPSN*, and *ZBTB14-EPB41L3* [23]. Recently, an MTAG combining SVS and WMH allowed for the identification of seven loci associated with SVS for the first time but some of them were previously associated with ICH (*SLC25A44-PMF1-BGLAP*, *LOX-ZNF474-LOC100505841*, *FOXF2 FOXQ1*, *SH3PXD2A*, *VTA1-GPR126*, *HTRA1-ARMS2*, and *COL4A2*) [23].

Proteome-wide association studies (PWAS) have been proposed as a novel strategy to recover causal protein-coding genes and to identify new associations for a disease. Since proteins are the focus of PWAS [24], it is a potentially beneficial tool for the identification of drug targets for disease [25]. This method, combined with Bayesian or Mendelian randomization, allows for the identification of gene-phenotype associations causally mediated by protein levels [24,25,26]. We performed a proteome-wide association study (PWAS) to integrate the genomic and proteomic data to discover common mechanisms regulating the two main acute manifestations of SVD: SVS and ICH.

## 2. Results

### 2.1. Stringent Criterion PWAS Analysis

From the 1475 proteins evaluated in the discovery cohort, only ICA1L reached a significant *q*-value in association with SVS (z-score = −4.93; *q*-value = 1.42 × 10^−2^) and with non-lobar ICH (z-score = −4.8; *q*-value = 2.33 × 10^−3^) (Table 1). The results from the Bayesian colocalization analysis showed a high posterior probability for the rs7582720 to be the causal variant for both the ICA1L levels and the significant GWAS traits (SVS and non-lobar ICH) (posterior probability 4 (PP4) = 0.9 for SVS and non-lobar ICH) (Table 1). The association of ICA1L with SVS and non-lobar ICH was the only one to reach the stringent criterion, and it was considered for the replication stage. Both associations resulted as significant in the replication analysis (the z-score = −3.55; *q*-value = 5.08 × 10^−3^, and PP4 = 0.99 for SVS and the z-score = −4.07, *q*-value = 6.19 × 10^−4^, and PP4 = 0.99 for non-lobar ICH) (Table 1).

We also tested, specifically, those proteins from loci associated previously with SVD. The PWAS aimed to interrogate 26 proteins corresponding to 13 different loci previously associated with ICH (4 loci) and/or SVS (12 loci). From these, only ICA1L was present among the proteins analysed in the PWAS.

### 2.2. SMR and Conditional Analysis

In order to ensure that the genetic association of ICA1L with SVS and non-lobar ICH was mediated by the cis-regulation of the ICA1L protein levels, SMR and a conditional analysis were performed. The SMR results indicated that the pQTL was associated with SVS (*p*-value = 3.66 × 10^−5^) and non-lobar ICH (*p*-value = 1.81 × 10^−5^) through the ICA1L protein expression regulation (Table 2, Figure 1). Similarly, the conditional analysis plot showed a conditional effect by the expression levels of ICA1L on the GWAS results (Figure 2). Table 2 shows the Summary statistics for the SMR results with the beta (b_SMR) and *p*-values (p_SMR) for all the significant and replicated associations from PWAS.

### 2.3. Open Criterion PWAS Analysis

In considering the open criterion (*p*-value < 1 ×10^−5^ in the discovery PWAS), five proteins reached a nominal association with all ICH, four proteins with SVS (excluding ICA1L) and lobar ICH, and two proteins with non-lobar ICH (excluding ICA1L) (Table 3). In the replication of the open analysis, only SPATA20 and ICA1L were significant (*p*-value < 0.05), with a PP4 higher than 0.5 in the replication proteome panel. SPATA20 was significantly associated with SVS and lobar ICH (*q*-value = 1.95 × 10^−3^ and PP4 = 0.97 for SVS; 8.82 × 10^−3^ and PP4 = 0.55 for lobar ICH). ICA1L was significantly associated with all ICH (*q*-value = 1.25 × 10^−2^ and PP4 = 0.99 for all ICH). Two proteins (NRBF2 and OSBPL11) could not be evaluated in the replication because they were not present in the panel (Table 3). Results from the SMR indicated that an ICA1L abundance was mediating the association through the pQTL and all ICH (*p*-value = 3.66 × 10^−5^). However, the SMR results for the SPATA20 were not significant (*p*-value = 3.69 × 10^−1^ for SVS and 7.98 × 10^−1^ for lobar ICH) (Table 2). The conditional analyses for both proteins showed that the genetic results were conditionally affected by a protein abundance (Figure 2 and Figure 3).

## 3. Discussion

In the present study, we performed a PWAS for the two main acute manifestations of SVD: SVS and ICH. We identified an association at the proteomic level of ICA1L with SVS and non-lobar ICH. Moreover, ICA1L was also associated with lobar ICH in our study. However, it was not a statistically significant association after a multivariable correction.

Although different loci have previously been found to be associated with ICH and SVS, how these genes contribute to their risk is not fully understood. One approach used to prioritize the likely causal gene within an associated locus with multiple genes is the use of a transcriptome-wide association study (TWAS) [27]. The TWAS from the SVS GWAS results found that the expression of *SLC25A44* in the arteries and the 2q33·2 locus (*CARF*, *FAM117B*, *ICA1L*, and *NBEAL1*) in the arteries and brain was causally associated with an SVS risk [23]. Very similar results regarding the 2q33·2 locus were previously obtained by Persyn et al. and Sargurupremraj et al. in their TWAS of WMH [16,17]. In contrast, the *ULK4* expression was negatively associated with an SVS risk in the arterial tissues, whole blood, and in the brain [23].

Our study implemented a similar strategy, focused on proteins, that allowed for a better characterization of a previously known association of ICA1L with non-lobar ICH, SVS [4,23], and related SVD traits, such as WMH [16,17] and fractional anisotropy (FA) [16]. Based on our results, genetics are responsible for the cis-regulation of an ICA1L abundance in brain tissue. Specifically, the top pQTL for ICA1L determines a higher abundance in ICA1L, which, in turn, reduces the risk for non-lobar ICH and SVS.

Although no previous PWAS has investigated the association of a protein abundance with SVS nor ICH, an ICA1L abundance was previously found to be associated with Alzheimer’s disease (AD) risk using a PWAS approach [25,28]. These studies used the same proteomic data as in our work and found that a higher expression of the ICA1L protein was linked with a decrease in AD risk [25,28]. Thus, both results for AD and SVD showed that elevated ICA1L levels are a protective factor. These AD studies, using single-cell RNA-sequencing, also identified enrichment in the ICA1L expression in glutamatergic excitatory neurons [25,28]. Excitatory processes involving glutamate are related tothe cell death features in stroke and ICH [29,30]. Furthermore, there is a clear relationship between AD and SVD. Both have common features that lead to dementia, such as a blood–brain barrier breakdown [31]. Moreover, cerebral amyloid angiopathy and arteriolosclerosis, common SVD mechanisms, worsen the threshold for AD dementia. Different SVD features have been found in post-mortem brains from AD patients, such as small cortical and subcortical infarcts, microbleeds, perivascular spacing, and WMH [32]. Additionally, *ApoE* is the major genetic factor associated both with lobar ICH and AD [33]. The Ica1l results suggest that this protein could also be commonly regulated in AD and SVD.

SPATA20 was also identified as a potential protein affecting SVS and lobar ICH. Previous genetic studies using a GWAS found polymorphisms in this gene associated with platelet distribution width [34], reticulocyte count [34], and subcortical volume [35]. However, pleiotropy was not found in the SMR analysis, indicating that different genetic regulations could be mediating the protein abundance and the trait risk.

The main limitation of our study is the number of proteins that we were able to evaluate. In this sense, only ICA1L was present in the proteomic panel from all proteins from loci previously associated with SVS or ICH. Moreover, we only had access to proteomic data from dPFC, and other brain regions could be relevant in SVD.

The results from this PWAS, combined with the previous results at the genomic and transcriptomic levels, highlight the relevance of ICA1L in the risk of SVD and specific SVD outcomes. As all these outcomes (SVS, non-lobar ICH, and lobar ICH) are common manifestations of SVD, the ICA1L regulation could be a common mechanism in SVD. Given that this regulation is also produced at the protein level and in brain tissue, future studies could focus on the investigation of ICA1L as a potential drug target.

## 4. Materials and Methods

### 4.1. Proteome-Wide Associations Study (PWAS)

We used FUSION software (Boston, MA, U.S.) [36] to estimate the association of brain protein levels with SVS and ICH. FUSION is implemented to identify associations between a functional phenotype (i.e., protein expression levels) and a GWAS phenotype. FUSION takes two inputs: 1) Precomputed functional weights, and 2) GWAS summary statistics unified to a reference single-nucleotide polymorphism (SNP) panel. In the PWAS, the functional weights consist of estimations of the strength of association between proteomic expression levels and the SNPs from the cis-locus of the protein (only considered in the heritable proteins). The SNPs associated with cis-locus protein expression levels are protein quantitative-traits (or pQTLs).

In our PWAS analysis, we selected, as functional weights, the freely available precomputed weights for the proteome expression analysis published by Wingo et al. [28]. As mentioned, in this case, the functional weights consisted of a summary-based file with data for the association between protein levels and SNPs (for the same locus of the protein). In the study from Wingo et al., from which we selected the functional weights, post-mortem dorsolateral prefrontal cortex (dPFC) human brain proteomes from 400 participants of European ancestry from the ROS/MAP cohorts were generated [37]. Proteomic sequencing was performed using isobaric tandem mass tag (TMT) peptide labelling combined with liquid chromatography coupled to mass spectrometry. They carried out different quality controls (QCs), which are described in the original article [28]. First, they randomized the 400 samples and 100 global internal standards into 50 batches before TMT labeling with the TMT10plex kit (Thermo Fisher Scientific, Waltham, MA, USA). Each batch contained two exact global internal standards. After running mass spectrometry, they assigned a peptide spectral using the Proteome Discoverer suite v.2.3 (Thermo Fisher Scientific) and a search in the canonical UniProtKB human proteome database. Then, the peptide spectral matches were collated into proteins. They did not include, in the analysis, the peptides whose protein quantification was outside the 95% confidence interval of the global internal standard, nor proteins with missing values in more than 50% of the samples. After the QCs, 8356 proteins remained for analysis. The abundance of these proteins was log2 transformed. Then, QC for the samples was also performed, removing samples outstanding more than four standard deviations from the first two principal components. The effect of the batch, sex, age of death, study, mass spectrometry reporter quantification mode, postmortem interval, and clinical diagnosis was detected and removed [28].

They coupled this proteomic data with genetic data for weight calculation purposes, and the genotype data from 1,190,321 HapMap SNPs present in 489 individuals from the same cohort was processed. The genotypes were obtained from whole-genome sequencing or genome-wide genotyping. Classical QCs were applied to the genotype data: removal of variants deviating from the HardyWeinberg equilibrium, with a missing genotype rate of >5%, and rare variants with a minor allele frequency of <1%. Individuals with >5% genotyping missingness and related individuals were also excluded from the analysis. For genome-wide genotyped data, imputation was performed, and only HapMap SNPs selected from the 1000 Genomes Project were included. After considering all the QCs for proteomic and genotyping data, the functional weight included dPFC proteome abundance from heritable proteins (a total of 1475 heritable proteins) in 376 subjects, combined with 1,190,321 SNPs [28].

For the PWAS, we selected the GWAS summary statistics from the traits of interest considering the largest sample size published. For SVS, we selected summary statistics for the European ancestry MEGASTROKE SVS GWAS (N = 5386 SVS patients and 406,111 controls) [22]. For ICH, we selected the three different ICH subtypes (all ICH, ICH lobar, and ICH non-lobar) GWAS summary statistics available from the MTAG published by Chung et al. (N = 6255 ICH cases and 233,058 control subjects) [4].

Using FUSION, we first unified the GWAS SNPs (total SNPs SVS = 8,177,651; total SNPs all ICH = 7,136,230; total SNPs non-lobar ICH = 7,136,333; and total SNPs lobar ICH = 7,495,490) with the reference panel from the 1000 Genomes Project (Linkage disequilibrium (LD) reference panel). Secondly, SNPs from the GWAS not present in the LD reference panel were imputed. Finally, the association between the functional weights (1475 proteins) and the GWAS summary statistics was estimated. We corrected the results, considering multiple testing by 1475 proteins with a False Discovery Rate (FDR) multivariable test correction (*q*-values).

### 4.2. Colocalization Analysis

We performed a Bayesian colocalization analysis for the PWAS results with a *p*-value of <0.05. The objective of this analysis was to assess whether the association of protein levels with the GWAS phenotype was caused by the same variant (or pQTL). We used COLOC, currently implemented in FUSION, for this purpose. Five different COLOC hypotheses were tested: no association (posterior probability (PP0), functional association only (PP1), GWAS association only (PP2), independent functional/GWAS associations (PP3), or colocalized functional/GWAS associations (PP4)). We considered plausible causal relationships for those with a posterior PP4 > 0.5.

We considered two criteria to assign significant terms to the PWAS results from the discovery analysis. First, using a stringent criterion, we considered significant PWAS results, those proteins with a significant *q*-value (<0.05) in the discovery PWAS, and a PP4 > 0.5 in the colocalization analysis [28]. Proteins following this criterion were considered in the replication analysis. Second, using an open criterion, we selected for the replication of all the proteins with a nominal *p*-value in the discovery PWAS (*p*-value < 10^−5^) (Figure 4).

### 4.3. Replication Analysis

For replication purposes, we selected a confirmation dPFC brain proteome panel obtained in 198 participants of European ancestry from the Banner Sun Health Research Institute [38]. Proteomic profiles were obtained using the same methodology used in the discovery panel [28]. After applying the QCs previously described, a total of 152 patients with proteomic and genetic data were used in the functional weight construction. Finally, proteomic weights included 1139 heritable proteins [28] that were combined with the above-mentioned GWAS summary statistics using the same protocol we used for the discovery panel.

All the proteins evaluated were considered significant in the replication study when the *q*-value was <0.05 (adjusting for all the proteins analysed in the replication) and the PP4 was >0.5 (Figure 1).

### 4.4. Summary-Based Mendelian Randomization (SMR)

We applied SMR to test whether the cis-regulated protein abundance was pleiotropically regulating the relationship between the SNPs and the trait of interest [39]. To perform the analysis for proteins achieving the stringent PWAS criterion, we followed the predetermined criteria of at least one pQTL at a *p*-value < 5 × 10^−8^ and included only SNPs with pQTL *p*-values < 1.5654 × 10^−3^. SNPs with LD r2 between the top SNP > 0.9 or <0.05 were excluded [39]. For the results from the open criterion, we included SNPs with pQTL *p*-values < 0.05.

We considered that protein levels were mediating the effect of an SNP on the phenotype when they had a *p*-value < 0.05 in SMR.

### 4.5. Conditional Analysis

For the significant PWAS proteins, we identified SNPs from the GWAS (*p*-value < 0.05) in the regions cis-overlapped with the selected proteins. In order to assess whether the significant associations between these polymorphisms and the trait were mediated by the cis-regulated protein abundance, we applied conditional tests using FUSION [27]. We plotted the results to observe that mediation visually, and we calculated conditional statistics. 

## Figures and Tables

**Figure 1 ijms-23-03161-f001:**
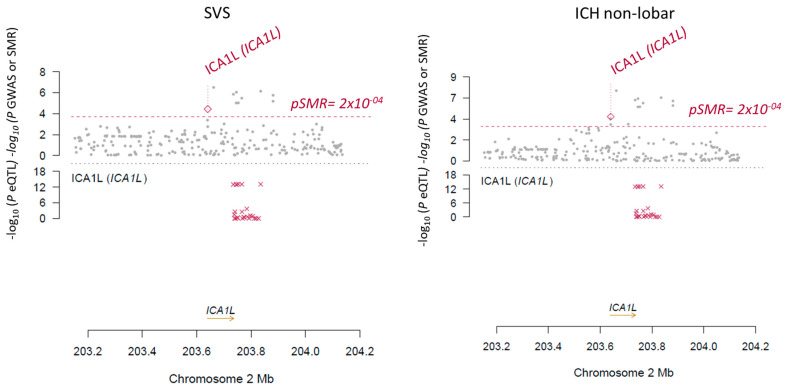
Significant SMR results for the significant and replicated PWAS results.Each “x” in figure represents a pQTL for ICA1L.

**Figure 2 ijms-23-03161-f002:**
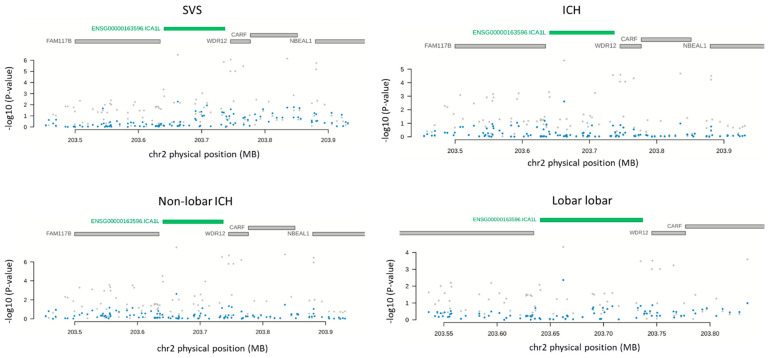
GWAS results before conditioning by ICA1L levels (grey) or after conditioning by protein levels (blue).

**Figure 3 ijms-23-03161-f003:**
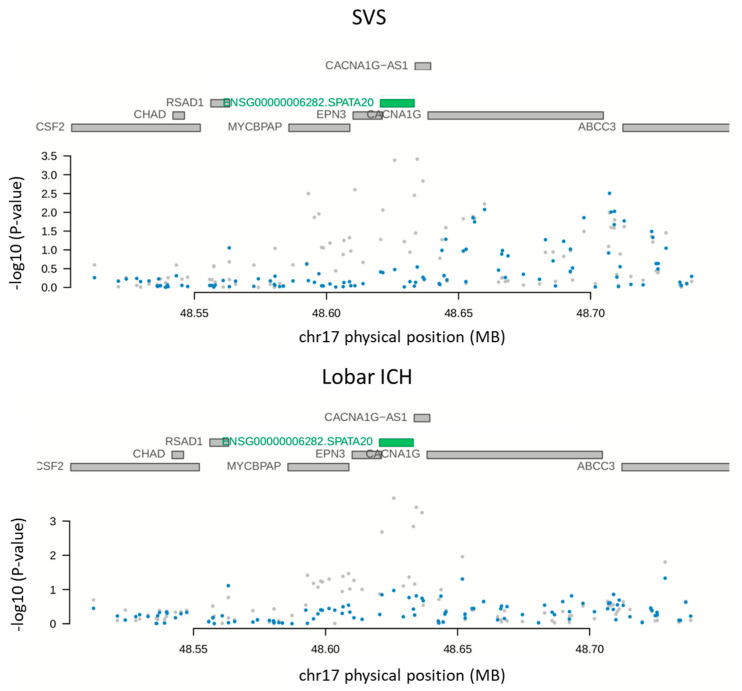
GWAS results before conditioning by the SPATA20 levels (grey) or after conditioning by protein levels (blue).

**Figure 4 ijms-23-03161-f004:**
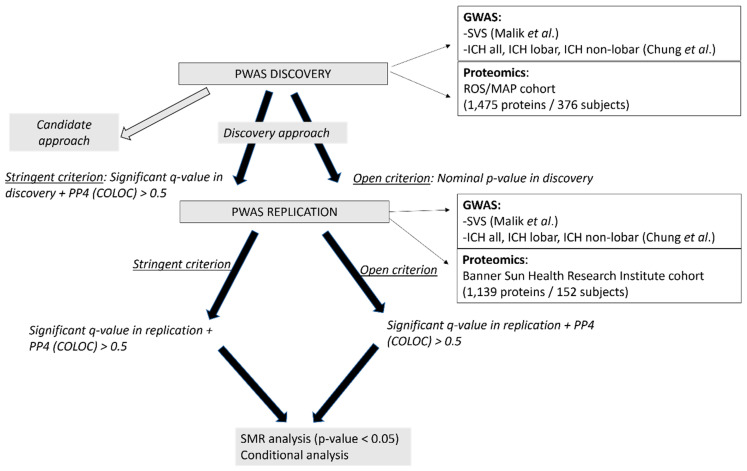
Schematic study workflow considering the discovery and the replication PWAS but also the sensitivity analyses.

**Table 1 ijms-23-03161-t001:** PWAS results for the stringent criteria. Significant PWAS associations using the stringent criteria in the discovery and replication. P0–P1: initial-final position for the analysed protein; pQTL ID: ID for the pQTL strongly associated with ICA1L; pQTL z-score: z-score for the pQTL strongly associated with ICA1L; pQTL z-score: z-score in the correspondent GWAS for the best pQTL; NSNP: number of SNPs in the locus; TWAS z-score: z-score for the PWAS association; PWAS *p*-value: *p*-value for the PWAS association; PP4: posterior probability 4 in the COLOC analysis; *q*-value: FDR-adjusted PWAS *p*-value.

				Discovery								Replication			
Trait	Protein	Chr	P0–P1	pQTL ID	pQTL z-Score	pQTL GWAS z-Score	NSNP	PWAS z-Score	PWAS *p*-Value	PP4	*q*-Value	PWAS z-Score	PWAS *p*-Value	PP4	*q*-Value
SVS	ICA1L	2	203640690 - 203736708	rs7582720	7.21	−4.426	81	−4.43	9.60 × 10^−6^	0.90	1.42 × 10^−2^	−3.55	3.91 × 10^−4^	0.99	5.08 × 10^−3^
Non-lobar ICH						−4.801		−4.80	1.58 × 10^−6^	0.93	2.33 × 10^−3^	−4.07	4.76 × 10^−5^	0.99	6.19 × 10^−4^

**Table 2 ijms-23-03161-t002:** SMR results.

Trait	Protein	b_SMR	p_SMR
SVS	ICA1L	−2.78	3.66 × 10^−5^
SVS	SPATA20	−0.24	3.69 × 10^−1^
ICH	ICA1L	−1.55	1.81 × 10^−5^
Non-lobar ICH	ICA1L	−1.55	1.81 × 10^−5^
Lobar ICH	SPATA20	−0.05	7.98 × 10^−1^

**Table 3 ijms-23-03161-t003:** PWAS results for the open criteria. The results for the discovery and replication of PWAS associations for proteins with a nominal *p*-value in the discovery (open criteria).

				Discovery							Replication			
Trait	Protein	Chr	pQTL ID	pQTL z-Score	pQTL GWAS z-Score	NSNP	PWAS z-Score	PWAS *p*-Value	PP4	*q*-Value	PWAS Z-Score	PWAS *p*-Value	PP4	*q*-Value
SVS	SPATA20	17	rs878619	−10.89	−2.917	102	3.71	2.10 × 10^−4^	0.17	3.10 × 10^−1^	3.79	1.50 × 10^−4^	0.97 *	1.95 × 10^−3^
SVS	ALDH2	12	rs4648328	−7.25	−3.437	61	3.58	3.38 × 10^−4^	0.29	4.99 × 10^−1^	3.38	7.28 × 10^−4^	0.27	9.46 × 10^−3^
SVS	EXOC6	10	rs980204	4.98	−2.469	210	−3.58	3.41 × 10^−4^	0.19	5.03 × 10^−1^	3.37	7.36 × 10^−1^	-	1
SVS	OSBPL11	3	rs2922170	4.11	3577	111	3.58	3.48 × 10^−4^	0.35	5.14 × 10^−1^	-	-	-	
ICH	ICA1L	2	rs7582720	7.21	−3.935	81	−3.94	8.32 × 10^−5^	0.68	1.23 × 10^−1^	−3.30	9.66 × 10^−4^	0.99 *	1.25 × 10^−2^
ICH	NRBF2	10	rs4379723	−4.19	−3.96	77	3.85	1.19 × 10^−4^	0.18	1.76 × 10^−1^	-	-	-	
ICH	DLGAP2	8	rs7842425	4.32	−3.742	225	−3.74	1.83 × 10^−4^	0.12	2.70 × 10^−1^	1.14	2.54 × 10^−1^	-	1
ICH	SPATA20	17	rs878619	−10.89	−2.89	102	3.58	3.47 × 10^−4^	0.12	5.12 × 10^−1^	3.27	1.09 × 10^−3^	0.33	1.41 × 10^−2^
ICH	MADD	11	rs11570115	−9.09	−2.565	99	3.54	4.06 × 10^−4^	0.03	5.99 × 10^−1^	1.88	6.04 × 10^−2^	-	0.78
Non-lobar ICH	SPATA20	17	rs878619	−10.89	−2.672	102	3.69	2.19 × 10^−4^	0.42	3.23 × 10^−1^	3.51	4.44 × 10^−4^	0.27	5.77 × 10^−3^
Non-lobar ICH	NRBF2	10	rs4379723	−4.19	−3.569	77	3.43	6.00× 10^−4^	0.29	8.90 × 10^−1^	-	-	-	
Lobar ICH	MRVI1	11	rs753002	−5.36	2.452	211	−3.69	2.20 × 10^−4^	0.02	3.25 × 10^−1^	−1.87	6.22× 10^−2^	-	0.80
Lobar ICH	NRBF2	10	rs4379723	−4.19	−3.804	77	3.68	2.33 × 10^−4^	0.11	3.44 × 10^−1^	-	-		
Lobar ICH	SPATA20	17	rs878619	−10.89	−3.186	102	3.51	4.44× 10^−4^	0.28	6.55 × 10^−1^	3.40	6.79× 10^−4^	0.55 *	8.82 × 10^−3^
Lobar ICH	ICA1L	2	rs7582720	7.21		81	−3.30	9.6× 10^−^^4^	0.2	1	−2.5692	1.02 × 10^−2^	0.9	0.13

* Significant PP4 in the replication. pQTL ID: ID for the pQTL strongly associated with proteins; pQTL z-score: z-score for the pQTL strongly associated with proteins; pQTL z-score: z-score in the correspondent GWAS for the best pQTL; NSNP: number of SNPs in the locus; TWAS z-score: z-score for the PWAS association; PWAS *p*-value: *p*-value for the PWAS association; PP4: posterior probability 4 in the COLOC analysis; *q*-value: FDR-adjusted PWAS *p*-value.

## Data Availability

The datasets used and/or analysed during the current study are available from the corresponding author upon reasonable request.

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
