# Peer review of "ICA1L Is Associated with Small Vessel Disease: A Proteome-Wide Association Study in Small Vessel Stroke and Intracerebral Haemorrhage"

_ijms, 2022, doi:10.3390/ijms23063161_

Round 1
Reviewer 1 Report
The materials and methods are short and mostly are cited without detailed explanation
beside ICA1L is seems that no other protein was identified significantly
Author Response
We appreciate your revision of our manuscript. We have included more detailed information in the Material and Methods section to clarify the methodology applied in our analysis.
We performed a PWAS to identify associations between brain protein levels and small vessel stroke (SVS) / intracerebral hemorrahges (ICH). With this purpose, we downloaded the needed data to run the PWAS with the FUSION software: 1) Proteomic data published by Wingo et al*., in the appropiated functional weight format from the ROS/MAP cohort and 2) GWAS summary statistics for the MEGASTROKE SVS GWAS (Malik et al.*) and the ICH GWAS (Chung et al.*).
More details about the obtention of the proteomic data and the quality controls performed by Wingo et al. have been included now in the manuscript.
Then, we ran a colocalization analysis (using COLOC) to assess whether the same causal genetic variant was associated with the phenotype (SVS/ICH) and the protein levels. This analysis was done for proteins achieving a p-value < 0.05 in the PWAS.
We considered for replication all the proteins achieving two different criteria: 1) Stringent: all the significant proteins in the PWAS (with FDR p-value, or q-value < 0.05) and with a posterior probability 4 in the colocalization analysis > 0.5; 2) Open: all the nominal proteins in the PWAS (p-value < 0.05). To replicate these results, we downloaded proteomic functional weights from the same manuscript (Wingo et al.) but a different cohort (the Banner Sun Health Research Institute).
Finally, for the replicated proteins, we applied a Summary-based Mendelian Randomization (SMR) test to know whether the association between the GWAS variant and the trait (SVS/ICH) was causally regulated by the protein abundance. We also assessed the mediation of the protein abundance on the GWAS results for the significant and replicated results using a condional analysis.
Only ICA1L was identified and confirmed in our PWAS to be associated with both SVS and ICH. Despite the association of SPATA20 was identified in the open criterion analysis, it was not significant following the stringent criterion nor in the SMR analysis, and thereby it would require further confirmation.
* references
- Wingo, A.P.; Liu, Y.; Gerasimov, E.S.; Gockley, J.; Logsdon, B.A.; Duong, D.M.; Dammer, E.B.; Robins, C.; Beach, T.G.; Reiman, E.M.; et al. Integrating Human Brain Proteomes with Genome-Wide Association Data Implicates New Proteins in Alzheimer’s Disease Pathogenesis. Nat. Genet. 2021, 53, 143–146, doi:10.1038/s41588-020-00773-z.
- Malik, R.; Chauhan, G.; Traylor, M.; Sargurupremraj, M.; Okada, Y.; Mishra, A.; Rutten-Jacobs, L.; Giese, A.-K.; van der Laan, S.W.; Gretarsdottir, S.; et al. Multiancestry Genome-Wide Association Study of 520,000 Subjects Identifies 32 Loci Associated with Stroke and Stroke Subtypes. Nat. Genet. 2018, 50, 524–537, doi:10.1038/s41588-018-0058-3.
-Chung, J.; Marini, S.; Pera, J.; Norrving, B.; Jimenez-Conde, J.; Roquer, J.; Fernandez-Cadenas, I.; Tirschwell, D.L.; Selim, M.; Brown, D.L.; et al. Genome-Wide Association Study of Cerebral Small Vessel Disease Reveals Established and Novel Loci. Brain 2019, 142, 3176–3189, doi:10.1093/brain/awz233.
Next, we attach the subsections from the Materials and Methods that we have updated (they are also included in the new revised version of the manuscript with change control):
4.1. Proteome-wide associations study (PWAS)
We used FUSION software [27] to estimate the association of brain protein levels with SVS and ICH. FUSION is implemented to identify associations between a functional phenotype (i.e. protein expression levels) and a GWAS phenotype. FUSION takes two inputs: 1) Precomputed functional weights; and 2) GWAS summary statistics unified to a reference single-nucleotide polymorphism (SNP) panel. In PWAS, the functional weights consist of estimations of the strength of association between proteomic expression levels and the SNPs from the cis-locus of the protein (only considered in heritable proteins). The SNPs associated with cis-locus protein expression levels are protein quantitative-traits (or pQTLs)
In our PWAS analysis, we selected as functional weights the freely available precomputed weights for the proteome expression analysis published by Wingo et al. [28]. As mentioned, in this case, the functional weights consisted of a summary-based file with data for the association between protein levels and SNPs (for the same locus that the protein). In the study from Wingo et al., from which we selected the functional weights, post-mortem dorsolateral prefrontal cortex (dPFC) human brain proteomes from 400 participants of European ancestry from the ROS/MAP cohorts were generated [29]. Proteomic sequencing was performed using isobaric tandem mass tag (TMT) peptide labelling combined with liquid chromatography coupled to mass spectrometry. They carried out different quality controls (QCs) which are described in the original article [28]. First, they randomized the 400 samples and 100 global internal standards into 50 batches before TMT labeling with TMT10plex kit (Thermo Fisher Scientific). Each batch contained two exact global internal standards. After running mass spectrometry, they assigned peptide spectral using the Proteome Discoverer suite v.2.3 (Thermo Fisher Scientific) and the search in the canonical UniProtKB human proteome database. Then, the peptide spectral matches were collated into proteins. They did not include in the analysis the peptides whose protein quantification was outside the 95% confidence interval of the global internal standard nor proteins with missing values in more than 50% of the samples. After the QCs, 8,356 proteins remained for analysis. The abundance of these proteins was log2 transformed. Then QC for samples were also performed, removing samples outstanding more than four standard deviations from the first two principal components. The effect of batch, sex, age of death, study, mass spectrometry reporter quantification mode, posmortem interval and clinical diagnosis was detected and removed.
They coupled this proteomic data with genetic data for weight calculation purposes, the genotype data from 1,190,321 HapMap SNPs present in 489 individuals from the same cohort was processed. The genotypes were obtained from whole-genome sequencing or genome-wide genotyping. Classical QCs were applied to the genotype data: removal of variants deviating from Hardy-Weinberg equilibrium, with missing genotype rate > 5% and rare variants with minor allele frequency < 1%.. Individuals with > 5% genotyping missingness and related individuals were also excluded from the analysis. For genome-wide genotyped data, imputation was performed and only HapMap SNPs selected from the 1000 Genomes Project were included. After considering all the QCs for proteomic and genotyping data, the functional weight included dPFC proteome abundance from heritable proteins (a total of 1,475 heritable proteins) in 376 subjects combined with 1,190,321 SNPs [28].
For the PWAS, we selected the GWAS summary statistics from the traits of interest considering the largest sample size published. For SVS, we selected summary statistics for the European-ancestry MEGASTROKE SVS GWAS (N = 5,386 SVS patients and 406,111 controls) [22]. For ICH, we selected the three different ICH subtypes (all ICH, ICH lobar and ICH non-lobar) GWAS summary statistics available from the MTAG published by Chung et al. (N = 6,255 ICH cases and 233,058 control subjects) [4].
Using FUSION, we first unified the GWAS SNPs (total SNPs SVS = 8,177,651; total SNPs all ICH = 7,136,230; total SNPs non-lobar ICH = 7,136,333; total SNPs lobar ICH = 7,495,490) with the reference panel from the 1000 Genomes Project (Linkage disequilibrium (LD) reference panel). Secondly, SNPs from GWAS not present in the LD reference panel were imputed. Finally, the association between the functional weights (1,475 proteins) and the GWAS summary statistics was estimated. We corrected the results considering multiple testing by 1,475 proteins with FDR multivariable test correction (q-values).
4.2. Colocalization analysis
We performed a Bayesian colocalization analysis for PWAS results with a p-value < 0.05. The objective of this analysis was to assess whether the association of protein levels with the GWAS phenotype was caused by the same variant (or pQTL). We used COLOC, currently implemented in FUSION, for this purpose. Five different COLOC hypotheses were tested: no association (posterior probability (PP0), functional association only (PP1), GWAS association only (PP2), independent functional/GWAS associations (PP3) or colocalized functional/GWAS associations (PP4)). We considered plausible causal relationships those with a posterior PP4 > 0.5.
We considered two criteria to assign significance terms to PWAS results from the discovery analysis. First, using a stringent criterion, we considered significant PWAS results those proteins with a significant q-value (< 0.05) in the discovery PWAS and a PP4 > 0.5 in the colocalization analysis [28]. Proteins following this criterion were considered in the replication analysis. Second, using an open criterion, we selected for replication all the proteins with a nominal p-value in the discovery PWAS (p-value < 10-05) (Figure 1).
4.3. Replication analysis
For replication purposes, we selected a confirmation dPFC brain proteome panel obtained in 198 participants of European ancestry from the Banner Sun Health Research Institute [30]. Proteomic profiles were obtained using the same methodology used in the Discovery panel [28]. After applying the QCs previously described, a total of 152 patients with proteomic and genetic data were used in the functional weight construction. Finally, proteomic weights included 1,139 heritable proteins [28] that were combined with the above mentioned GWAS summary statistics using the same protocol we used for the discovery panel.
All the proteins evaluated were considered significant in the replication study when the q-value was < 0.05 (adjusting for all the proteins analysed in the replication) and the PP4 was > 0.5 (Figure 1).

Reviewer 2 Report
In the submitted manuscript, authors have made a decent attempt to show that association of ICAL1 with SVS and non lobar ICH is conditioned by cis-regulation of it's protein levels in the brain. The manuscript is well written and easy to follow through for readers. The topic is also of interest to readers working in the field and can provide significant advancement to current knowledge. I would really like to see in subsequent studies from the persons working in the field to provide some proof of concept studies targeting ICAL1 as it is involved in both SVD and AD. The quality of Figure 1 is not good. I would request the authors to provide Figure 1 with better resolution.
Author Response
Thank you very much for your comment. We consider that the association of ICA1L with both SVD and AD could be revealing a common pathophysiological mechanism between both diseases and we believe it can has future therapeutic implications. For that reason, we will continue this research line to investigate with functional analysis the mechanistic association of ICA1L with both SVD and AD.
We have updated the Figure 1 in order to improve its quality.

Round 2
Reviewer 2 Report
Thank you for replacing the Figure 1.